EpiCurator: an immunoinformatic workflow to predict and prioritize SARS-CoV-2 epitopes

Ferreira Cristina S. 1
Martins Yasmmin C. 1
Souza Rangel Celso 1
Vasconcelos Ana Tereza R. atrv@lncc.br 1
Bioinformatics Laboratory, National Laboratory of Scientific Computation , Petrópolis , Rio de Janeiro , Brazil
Lefkowitz Elliot
Electronic publication date: 2021 Nov 30
Publication date: 2021
Volume: 9
Electronic Location ID: e12548
Received 2021 Jul 3; Accepted 2021 Nov 4
Copyright: ©2021 Ferreira et al.
Copyright year: 2021
Copyright holder: Ferreira et al.
License: This is an open access article distributed under the terms of the Creative Commons Attribution License, which permits unrestricted use, distribution, reproduction and adaptation in any medium and for any purpose provided that it is properly attributed. For attribution, the original author(s), title, publication source (PeerJ) and either DOI or URL of the article must be cited.
License URL: https://creativecommons.org/licenses/by/4.0/

Keywords: Epitopes prediction, SARS-CoV-2, HLA alleles binding affinity, Immunoinformatic approaches, EpiCurator

Funding: Conselho Nacional de Desenvolvimento Científico e Tecnológico - CNPq 303170/2017-4 FAPERJ E -26/202.826/2018 E-26/202.168/2020 This work was developed in the frameworks of Corona-ômica-RJ (FAPERJ = E-26/210.179/2020) and Rede Corona-ômica BR MCTI/FINEP (FINEP = 01.20.0029.000462/20, CNPq = 404096/2020-4) and BRICS/CNPq - (440931/2020-7). Ana Tereza R. Vasconcelos is supported by Conselho Nacional de Desenvolvimento Científico e Tecnológico - CNPq (303170/2017-4) and FAPERJ (E -26/202.826/2018). Yasmmin C Martins is currently supported by FAPERJ (E-26/202.168/2020). The funders had no role in study design, data collection and analysis, decision to publish, or preparation of the manuscript.

==============================
The ongoing coronavirus 2019 (COVID-19) pandemic, triggered by the emerging SARS-CoV-2 virus, represents a global public health challenge. Therefore, the development of effective vaccines is an urgent need to prevent and control virus spread. One of the vaccine production strategies uses the in silico epitope prediction from the virus genome by immunoinformatic approaches, which assist in selecting candidate epitopes for in vitro and clinical trials research. This study introduces the EpiCurator workflow to predict and prioritize epitopes from SARS-CoV-2 genomes by combining a series of computational filtering tools. To validate the workflow effectiveness, SARS-CoV-2 genomes retrieved from the GISAID database were analyzed. We identified 11 epitopes in the receptor-binding domain (RBD) of Spike glycoprotein, an important antigenic determinant, not previously described in the literature or published on the Immune Epitope Database (IEDB). Interestingly, these epitopes have a combination of important properties: recognized in sequences of the current variants of concern, present high antigenicity, conservancy, and broad population coverage. The RBD epitopes were the source for a multi-epitope design to in silico validation of their immunogenic potential. The multi-epitope overall quality was computationally validated, endorsing its efficiency to trigger an effective immune response since it has stability, high antigenicity and strong interactions with Toll-Like Receptors (TLR). Taken together, the findings in the current study demonstrated the efficacy of the workflow for epitopes discovery, providing target candidates for immunogen development.

Introduction

The emergence of the SARS-CoV-2 infection, causing COVID-19 disease, has spread rapidly worldwide and represents a global challenge for public health (Cohen & Normile, 2020; Chakraborty et al., 2020). This virus was first reported in Wuhan in December 2019 (Huang et al., 2020) and quickly evolved, with the emergence of several variants (Cella et al., 2021; Brüssow, 2021). According to the World Health Organization (WHO) classifications, there are four variants of concern (VOCs) currently spread worldwide designated as Alpha (B.1.1.7), Beta (B.1.351), Gamma (P.1), and Delta (B.1.617.2) (Faria et al., 2020; Rambaut et al., 2020; WHO, 2021; Tegally et al., 2021). In addition, the VOCs are characterized by the possibility to impact the disease severity, such as the possible increased risk of hospitalization, mortality, and capability of evading vaccination-induced immune response (Prévost & Finzi, 2021; Geers et al., 2021). The constant emergence of new variants keeps the contagiousness of SARS-CoV-2, which increases the uncertainty of virus spread (Brüssow, 2021; Naveca et al., 2021).

Current research aims to develop effective interventions for controlling and preventing the COVID-19 pandemic; furthermore, vaccination is still the most economical and effective approach to prevent infection by the virus (Shang et al., 2020). At the time of writing, 22 vaccines against SARS-CoV-2 have been approved by at least one country (McGill COVID19 Vaccine Tracker Team, 2021). Five vaccines were approved for emergency use authorization and listed by WHO Emergency Use Listing (EUL) (Bulla, 2021; Mascellino et al., 2021). The Pfizer/BioNTech vaccine is based on messenger RNA (mRNA), coding for viral spike (S) protein (Badiani et al., 2020). The Moderna vaccine is a lipid nano-particle-encapsulated mRNA-based vaccine that encodes a full-length spike (Mahase, 2020). The Johnson & Johnson (Janssen) is a recombinant, non-replicating adenovirus vector encoding a full-length S protein (Livingston, Malani & Creech, 2021). The Oxford-AstraZeneca is a chimpanzee adenovirus vectored DNA vaccine (Knoll & Wonodi, 2021), and the CoronaVac is an inactivated SARS-CoV-2 virus vaccine (Gao et al., 2020; Mallapaty, 2021).

The current vaccines are generally based on B cell immunity with neutralizing antibody production (Thanh Le et al., 2020; Siracusano, Pastori & Lopalco, 2020; Yoshida et al., 2021). However, several studies report the capability of a new mutation interrupting the binding with some neutralizing antibodies (Greaney et al., 2021; Hoffmann et al., 2021; Andreano et al., 2021), which could clarify the spread of the current VOCs. The main reasons for this capability are the several changes (mutation and deletion) in the S protein, which is responsible by the increased affinity between the receptor-binding domain (RBD) and the human cellular receptor angiotensin-converting enzyme 2 (ACE2), promoting the antibodies escape (Thomson et al., 2021; Greaney et al., 2021; Liu et al., 2021; Rotondo et al., 2021).

To improve the vaccine’s effectiveness against SARS-CoV-2, global efforts in resources, cooperation, and innovation have contributed to the accelerated development of COVID-19 vaccines (Bloom et al., 2021). These efforts lead various researchers to carry out diversified methodologies of vaccine design such as a peptide-based vaccine, virus-like particle, replicating and non-replicating viral vectors, DNA or RNA, live attenuated virus, recombinant designed proteins, nanoparticles vaccine, and inactivated virus (Medhi et al., 2020; Krammer, 2020; Di Natale et al., 2020; Kyriakidis et al., 2021; Shahcheraghi et al., 2021).

This work highlights the peptide-based vaccine described in several studies about SARS-CoV-2 vaccines (Malonis, Lai & Vergnolle, 2020; Chakraborty et al., 2020; Crooke et al., 2020; Fatoba et al., 2021; Naveed et al., 2021). Currently, ten from 33 vaccines in phase 3 clinical trial, and four from 22 approved vaccines have a peptide-based design (McGill COVID19 Vaccine Tracker Team, 2021). The advantages of peptide-based vaccines include their capability to target very specific epitopes decreasing the risks associated with allergic and autoimmune responses, besides involving minimal viral components to stimulate adaptive immunity (Di Natale et al., 2020). Additionally, their chemical or recombinant cloning synthesis allows large-scale production with low costs and high reproducibility (Sun, 2013; Skwarczynski & Toth, 2016; Hudu, Shinkafi & Umar, 2016).

The peptide-based vaccine design requires the immunoinformatic approach as part of the computational vaccinology strategy for epitopes prediction (Ramana & Mehla, 2020; Oli et al., 2020; Lu et al., 2021). This approach regards the wide availability of the SARS-CoV-2 NGS (Next-Generation Sequencing) information associated with human leukocyte antigen (HLA) profile (Kazi et al., 2018; Oli et al., 2020; Sharma et al., 2020) to identify T cell epitopes. Therefore, these epitopes have the capability to effectively bind to HLA molecules activating a long-lasting immune response mediated by CD8+ and CD4+ T cells (Fast, Altman & Chen, 2020; Wang & Gui, 2020).

T cell epitopes offer advantages for vaccine design since it does not depend on the recognition of structural proteins (Bashir et al., 2021; Redd et al., 2021) and are less affected by deletions and mutations of emergent variants (Ribes, Chaccour & Moncunill, 2021; Jin et al., 2021). Concerning COVID-19 immune response, T cell epitopes have the potential to provide long-term protection from SARS-CoV-2. This characteristic allows the detection of memory T cell responses to multiple SARS-CoV-2 proteins, which might contribute to disease control (Chen & John Wherry, 2020; Karlsson, Humbert & Buggert, 2020; Sette & Crotty, 2021).

To provide effective T cell epitopes for in vitro peptide-based vaccine design a robust, refined and accurate in silico selection of epitopes is crucial. Despite the many immunoinformatic tools for epitope prediction, the curation for epitope selection is still limited and needs different web servers to complete the analysis. In this paper, we focus on a computational prediction, curation and validation of SARS-CoV-2 epitopes. We have as a central proposal a workflow (EpiCurator) that brings together different approaches for accurate selection of epitopes, providing the refined identification of promising SARS-CoV-2 epitopes. To validate the efficacy of this new tool, we use samples of circulating Brazilian lineages available in GISAID (https://www.gisaid.org/) (Elbe & Buckland-Merrett, 2017) from December 2020 to April 2021.

Materials & Methods

Genome retrieval and protein annotation

Genome sequences of 1,652 SARS-CoV-2 genome isolated in Brazil were retrieved from the GISAID database (Elbe & Buckland-Merrett, 2017) available from December 2020 to April 2021 (Table S1). The lineage distribution of the retrieved genomes includes P.1 (Gamma, n = 770), P.2 (Zeta, n = 525), B.1.1.28 (n = 223) and B.1.1.33 (n = 136). These genomes were analyzed by The Viral Annotation Pipeline and iDentification (VAPiD) (Shean et al., 2019) to determine the amino acid sequence for all SARS-CoV-2 proteins using as reference NC_045512.2 from the NCBI database (https://www.ncbi.nlm.nih.gov/). The final genome processing includes clustering the amino acid sequences for each protein using the CD-HIT package (Li & Godzik, 2006) with a 100% sequence identity threshold.

Spike protein comparison

Clustal Omega and MView tool performed a homology analysis for the Spike protein (Sievers & Higgins, 2018); (Brown, Leroy & Sander, 1998) among 100 random samples of Brazilian lineages and the VOCs, retrieved from the GISAID database (Elbe & Buckland-Merrett, 2017), to measure the identity of the sequences to Gamma (P.1) samples.

Epitope prediction

The T and B cell epitopes prediction was performed using the softwares NetCTL, NetMHCpan, and NetMHCIIpan (https://services.healthtech.dtu.dk/) that allow the high-throughput computing analysis. These predictors support FASTA files containing amino acids marked with “X” in the sequence expanding the possibility of public genomes analysis, even with minor sequencing errors. This prediction features decreasing the pre-processing sequence steps and avoids the false-positive epitopes provided by joining sequences.

Prediction of SARS-CoV-2 epitopes

To predict CD8+ T cell epitopes, the FASTA sequences of SARS-CoV-2 proteins are processed using NetCTL v1.2 (Larsen et al., 2007). First, the sequences and the HLA class I supertypes, provided by the software, are submitted to select 9-mer peptides (Chen et al., 1994; Sakaguchi et al., 1997; Gfeller et al., 2018), then peptides with amino acids marked with “X” are removed (Fig. 1A). The predicted peptides in NetCTL are further processed using NetMHCpan v.4.0 software (Hoof et al., 2009; Jurtz et al., 2017) to identify epitopes with strong binding affinity to HLA class I alleles. The prediction parameter is based on the predicted percentile rank ≤ 0.5% and half-maximal inhibitory concentration (IC50) < 500 nM (Chen et al., 1994; Sakaguchi et al., 1997; Gfeller et al., 2018) (Fig. 1A). To assess binding affinity, alleles of HLA-A, B, and C loci were selected from Allele Frequency Net Database (Gonzalez-Galarza et al., 2020) by their Brazilian population frequency > 5% (Table S2).

Figure 1 Schematic overview of the prediction, accurate selection and epitopes validation to identify new SARS-CoV-2 epitopes.

The pipeline comprises four main analyses: prediction and curation epitopes selection, immune properties evaluation and multi-epitope validation. (A) The prediction encompasses four DTU Health Tech softwares (blue boxes) subdivided according to the type of identified epitopes: CD8+ T cell, CD4+ T cell and B cell epitopes. (B) The curation step uses the predicted epitopes for an accurate selection comprising our proposal EpiCurator workflow (pink boxes) and online server to evaluate the antigenicity, toxicity and allergen analysis. (C) The pipeline has a set of individual analyses that identify the population coverage, immunogenicity and other immunogenic properties using on-line servers (yellow boxes). (D) Additionally, a in silico validation of final RBD selected epitopes has been performed using a multi-epitope construct. Their linear, 2D and 3D sequences are evaluated using online servers (yellow boxes) to characterize the multi-epitope and the construct-TLR complexes as stable and immunogenic. Each analysis has its respective parameters presented below the boxes.

To predict CD4+ T cell epitopes and estimate binding affinity to HLA class II molecules, the FASTA sequences are processed using NetMHCIIpan v3.2 software (Greenbaum et al., 2011). This tool selects the epitopes with 15-mer peptide lengths based on the predicted percentile rank ≤ 2.0% and IC50 < 500 nM (Fig. 1A). For the affinity prediction, the loci HLA-DRB1, HLA-DPA1-DPB1, and HLA-DQA1-DQB1 were selected by the phenotypic frequency > 5% in the Brazilian population (Table S2) from Allele Frequency Net Database (Gonzalez-Galarza et al., 2020). For both cells (CD8+ and CD4+ T cell), the epitopes with broad HLA affinity coverage (≥3 alleles) are filtered (Fig. 1A).

The prediction of linear B cell epitopes from SARS-CoV-2 structural proteins was performed by BepiPred v.2.0 software (Jespersen et al., 2017), with a threshold of 0.5 (corresponding specificity > 0.817 and sensitivity < 0.292) (Fig. 1A). Only the epitopes with more than seven and less than 50 amino acid residues in length and sequence without amino acids marked with “X” are considered for subsequent curation analysis.

Accurate selection of epitopes - EpiCurator

We developed a rigorous analysis workflow for accurate selection of epitopes, the EpiCurator, bringing together a set of filters according to pre-established criteria, optimizing the analysis since it brings tools available to use by high-throughput computing architectures. They also group a series of analyses to guarantee the selection of unpublished and qualified epitopes. The workflow allows the identification of epitopes by the following analysis: conservancy, homology with the human genome, the overlap between epitopes of different classes, and the identification of epitopes previously published in PubMed Central® (PMC) or available in the IEDB database (Vita et al., 2019) beyond identifying their protein coordinates and mutations (Fig. 1B).

Prediction of epitope conservancy

The first module of EpiCurator calculates the conservancy of a predicted epitopes list in SARS-CoV-2 genomes using the BLAST command-line tools (Madden, 2020) (Fig. 1B). A customized BLAST database, optimized for shorter sequences (blastp-short task), was generated with proteins of some SARS-CoV-2 circulating lineages from Brazil (B.1.1.28, P.1, P.2) retrieved from the GISAID database (2,787 genome sequences) (Elbe & Buckland-Merrett, 2017). This database allows the comparison among the sequences of the predicted epitopes and the SARS-CoV-2 proteins. Thus, the analysis reports the percentage of identity (conservancy) per epitope using four criteria: 100% of identity; above 90%; between 70% and 90%, and less than 70%. The epitopes conserved with 100% of identity in at least 90% of the genomes are used for further analysis (Fig. 1B).

Human homology

This module uses the human proteins dataset from Ensembl (GRCh38.p13) to identify the predicted epitopes sequence in the human genome. The workflow keeps the human protein sequences in memory to enhance the analysis readiness, searching strictly for the corresponding epitope sequences (exact match). Only the unmatched epitopes are selected, returning a filtered list of epitopes without human homology (Fig. 1B).

Epitope sequence overlap

A comparison is performed between the sequence of the epitopes derived from three groups of prediction (B Cell, HLA Class I, and HLA Class II T Cell). This analysis calculates the intersection and the identity among the lists of epitopes belonging to these groups returning four reports: (i) the intersection of all groups, (ii) B cell x Class I epitopes, B cell x Class II epitopes, and (iii) Class I x Class II epitopes. The result keeps the epitopes with less than 60% similarity in each report (Fig. 1B).

Search for epitopes from published articles - EpiMiner

The EpiMiner was developed to execute an automatic search of the predicted epitopes list on Pubmed and PMC published papers (Canese & Weis, 2013) (Fig. 1B). It is a pipeline performed in four steps. The first retrieves Pubmed and PMC articles that include the epitopes sequence. The second step extracts the sentences of the body, abstract or tables of the article, highlights the figure captions and breaks apart supplementary table data. The third step executes natural language processing techniques such as tokenization to divide the sentences into words and part-of-speech tagging to filter nouns and verbs (Chowdhury, 2005). These techniques contribute to reducing search time for epitope sequence recognition, eliminating uninformative sentences. The fourth step executes an entity recognition to identify epitopes sequence classified as nouns, saving the sentences and their respective publication into a report. Epitopes information is not searched if it is a part of the pictures or is in supplementary file in text format (i.e., .doc, .docx, .txt, .pdf).

IEDB matching

To perform this analysis, the complete ensemble of epitopes from the Immune Epitope Database (IEDB) v3 release (https://www.iedb.org/database_export_v3.php) was retrieved (accessed on July 2021), and the structured files were parsed to filter the epitopes belonging to SARS-CoV-2 organisms (n = 1, 268) (Vita et al., 2019). This filtered file is used to calculate the identity between the predicted list of epitopes and the ones retrieved, apprising the IEDB epitope ID and the respective similarity (Fig. 1B).

Mutation screening

Epitopes mutation is reported by comparing the epitope sequence with the SARS-CoV-2 Wuhan protein sequence and assigning the epitopes’ coordinates in the respective whole protein. The alignment is conducted with the blastp function of the BLAST command-line tools (Madden, 2020). The SARS-CoV-2 reference protein sequences used for alignment are published on the Uniprot Database (https://covid-19.uniprot.org/). This analysis describes the coordinates and presence of mutations in the predicted epitopes (Fig. 1B).

Epitope properties

Evaluation of antigenicity, toxicity, and immunogenic profile

The VaxiJen v2.0 server was applied to analyze the antigenicity of the predicted B cell and T cell epitopes with a conservative score threshold of 0.7 (Doytchinova & Flower, 2007a; Doytchinova & Flower, 2007b) (Fig. 1B). The toxicity is retrieved from the ToxinPred online server with support vector machine (SVM) based methods (threshold −0.4) and e-value cut-off 0.01 (Gupta et al., 2013) (Fig. 1B). The allergenic properties are retrieved from the AlgPred2 online server with a hybrid prediction model (threshold 0.5) (Sharma et al., 2020) (Fig. 1B). Despite being part of the accurate selection (Fig. 1B), they are not available for incorporation into the analytic workflow (EpiCurator) which prevents us from optimizing their assay. In addition, several online servers were used to evaluate the immunogenic profile of the T cell epitopes (Fig. 1C). The immunogenicity predictions were performed by the IEDB server (https://www.iedb.org/) (Paul et al., 2015; Calis et al., 2013; Dhanda et al., 2018; Vita et al., 2019). Likewise, their capability to induce interferon-gamma (IFNγ), interleukin-4 (IL-4), interleukin-10 (IL-10), and interleukin-17 (IL-17) (Dhanda, Vir & Raghava, 2013; Dhanda et al., 2013; Gupta et al., 2017; Nagpal et al., 2017) was evaluated. Additionally, their proinflammatory activity (Gupta et al., 2016) and immunomodulatory potential (Nagpal et al., 2018) was verified. The servers and parameters to evaluate the immunogenic profile are shown in Fig. 1C.

Estimation of population coverage

The IEDB’s Population Coverage online tool (http://tools.iedb.org/population/) is used to analyze how T cell epitopes-HLA binding alleles diverge across ethnicities, regions, and countries around the world (Bui et al., 2006; Vita et al., 2019) (Fig. 1C). The predicted epitopes and their respective HLA binding alleles (Table S3) were inputted in the IEDB tool with separated allele class option and a list of countries/regions (Argentina, Brazil, England, France, Italy, Spain, United States, and World) was selected.

Docking analysis of the HLA-epitope complex

To validate the binding affinity of the predicted epitopes with HLAs structures, docking analysis was performed using the PepDock tool (Lee et al., 2015) of the GALAXY web server (Fig. 1C). To docking analysis, the Protein Data Bank archive (PDB) of eight HLA alleles (HLA-A*01:01, HLA-A*02:01, HLA-B*08:01, HLA-C*12:03, HLA-DRB1*03:01, HLA-DRB1*04:01, HLA-DRB1*12:02 and HLA-DRB1*15:01) was retrieved from the pHLA3D database (Menezes Teles et al., 2019) and RCSB PDB database (Berman et al., 2000; Burley et al., 2021).

The PepDock uses the HLA alleles PDBs and the epitopes sequence to perform a template-based model selection and then proceeds to the docking and refinement processes to optimize the energy score ranking ten complex models. The best model of each HLA-epitope complex was selected based on the major similarity score of protein structure and interaction beyond the highest estimated accuracy. Additionally, to identify the free energy and the residue’s contacts of the selected complexes, the Prodigy tool (Xue et al., 2016) and Chimera software (Goddard, Huang & Ferrin, 2005) were used respectively.

Genomes for EpiCurator pairwise comparison validation

Protein sequences for SARS-CoV-2 isolates reported by Crooke et al. (2020) were identified and retrieved from the Virus Pathogen Resource (ViPR) database (n = 641, 635); additionally, six genome sequences reported by Kiyotani et al. (2020), two sequences for N and S protein reported by Chen et al. (2020), and five S sequences reported by Chukwudozie et al. (2021) were retrieved from NCBI GenBank. These samples were processed using prediction parameters of the HLA alleles reported by the authors with the approach reported in this Methods section (Epitope prediction and Accurate selection analysis). Regarding item Accurate selection of epitopes - EpiCurator of the Methods, we used only the three main analysis steps (conservancy, human homology and IEDB matching) to compare the selected epitopes ensemble by the papers, and ones chosen independently by our approach, leaving out the EpiMiner analysis since all the epitopes are published.

Multi-epitope construct and structural modelling for EpiCurator validation

The RBD epitopes (n = 11) were used to construct a multi-epitope sequence connected by specific linkers (Fig. S1). The linker aimed to separate the epitopes, so that it improved their expression, folding and stability beyond to prevent their fusion and facilitate the immune processing of antigen (Arai et al., 2001; Kar et al., 2020). In the multi-epitope arrangement are also added an adjuvant (Escherichia coli 50S ribosomal protein L7/L12 (UniProt P0A7K2)) in the N-terminal sequence and a histidine hexamer in the C-terminal portion (Fig. 1D, Fig. S1).

Linear and secondary structure evaluation

To evaluate the properties and immune profile of multi-epitope, its linear sequence was submitted to several analyses (Fig. 1D) as follow: antigenicity (Doytchinova & Flower, 2007a; Magnan et al., 2010), allergenicity (Dimitrov et al., 2014b; Dimitrov et al., 2014a), and solubility for cell-free expression analysis (Hebditch et al., 2017), for overexpression analysis (Magnan, Randall & Baldi, 2009), and structurally solubility profile (Hou et al., 2020). In addition, we assess its physicochemical properties (Gasteiger et al., 2005). The servers and parameters to linear multi-epitope properties evaluation are shown in Fig. 1D.

The linear sequence was also analyzed by the C-IMMSIM server (https://kraken.iac.rm.cnr.it/C-IMMSIM/) to evaluate the in silico immune profile of the multi-epitope construct (Rapin et al., 2010). Two simulations were performed with intervals of 4 or 12 weeks (Saad-Roy et al., 2021; Cobey et al., 2021) (Fig. 1D). Furthermore, the secondary structure of the multi-epitope was evaluated with PSIPRED v.4 with an accuracy of 84.2% (Jones, 1999; McGuffin, Bryson & Jones, 2000; Buchan & Jones, 2019) (Fig. 1D).

Multi-epitope 3D structure modelling, refinement, and evaluation

The tertiary structure multi-epitope construct is modeled by the RaptorX server (http://raptorx.uchicago.edu/ContactMap) that predicts structural properties such as solvent accessibility (ACC) and disorder regions (DIS) (Källberg et al., 2014) (Fig. 1D). This tertiary structure was submitted to a refinement process using the GalaxyRefine (http://galaxy.seoklab.org/cgi-bin/submit.cgi?type=REFINE) server which improves global and local model quality by rebuilding all side-chain conformations and applying structural relaxations (Heo, Park & Seok, 2013). The quality of the refined model is assessed by several parameters, as reported in Fig. 1D. The refined 3D structure is further evaluated by the ProSA-web server (https://prosa.services.came.sbg.ac.at/prosa.php) to validate the structural models checking for potential errors (Berman et al., 2000; Wiederstein & Sippl, 2007) (Fig. 1D). The final refined 3D structure was used to perform a docking by the ClusPro server (https://cluspro.bu.edu) (Desta et al., 2020) using the multi-epitope 3D structure and the Toll-Like Receptors TLR4 (PDB: 2Z63) and the TLR3 (PDB: 3CIG). The complex model with the lowest energy is chosen for each receptor (Fig. 1D). Their stability is assessed with molecular dynamics simulation on the iMODS server (http://imods.chaconlab.org) that performs Normal Mode Analysis (NMA) to describe functional motions between macromolecules in complexes and simulates feasible trajectories between two conformations (López-Blanco et al., 2014) (Fig. 1D).

Results

Linear epitopes prediction from SARS-CoV-2 genome

Three main epitopes prediction comprise HLA Class I (CD8+ T cell), HLA Class II (CD4+ T cell), and B cell. To predict potential CD8+ T cell epitopes, the NetCTL and NetMHCpan, predictive algorithms were performed for all proteins annotated from 1,652 SARS-CoV-2 genomes (see Methods). Thus 9-mer epitopes were predicted with strong binding affinity assigned by percentile scores (rank) ≤0.5% across HLA class I alleles frequent in the Brazilian population (Fig. 1A). Cumulatively reaching the prediction of 5,261 HLA class I epitopes with HLA promiscuity, binding to ≥3 HLA alleles, implying broad population coverage. The affinity to HLA-C alleles matches the highest epitope binding locus suggesting it could be the best grooves for these epitopes and would lead to the activation of the cell-mediated response (Table 1).

Table 1 Summary of SARS-CoV-2-derived epitopes predicted with high binding affinity to each HLA loci.

Protein	Protein length
(amino acids)	Number of epitopes	
		HLA class I-restricted T cell epitopes (rank ≤ 0.5)	HLA class II-restricted T cell epitopes (rank ≤ 2)	B cell epitopes	
		All	HLA-A	HLA-B	HLA-C	All	HLA-DP	HLA-DQ	HLA-DR	All	
Envelope	75	33	22	26	69	54	–	–	13	10	
Membrane	222	99	104	55	127	83	6	6	44	7	
Nucleocapsid	419	226	143	205	217	390	21	22	202	147	
ORF10	38	23	12	12	29	10	–	1	8	–	
ORF1ab	7096	3621	2454	3517	4262	5225	1291	630	2608	–	
ORF3a	275	343	256	298	435	357	46	24	260	–	
ORF6	61	32	15	20	26	47	13	4	24	–	
ORF7a	121	96	65	110	96	102	101	2	71	–	
ORF7b	43	15	9	7	15	3	–	–	1	–	
ORF8	121	98	51	87	113	139	57	8	35	–	
Spike	1273	675	427	652	775	1239	531	120	436	93	
TOTAL	-	5261	3558	4989	6164	7649	2066	817	3702	257	

We also sought to predict potential 15-mer epitopes with binding affinity to HLA class II, using the NetMHCIIpan software (Fig. 1A), reaching 7,649 candidate HLA class II epitopes from the SARS-CoV-2 genomes. They have a strong binding affinity to ≥3 HLA alleles across a reference panel of HLA molecules (see Methods). These epitopes showed a preferential affinity for alleles from locus HLA-DRB1 (n = 3, 702), suggesting that HLA-DRB1 alleles could lead these epitopes to activate the cellular immune response (Table 1).

To pairwise the cellular and humoral immune responses activation, the additional prediction of linear B cell epitopes was performed by BepiPred algorithm identifying 257 epitopes from SARS-CoV-2 structural proteins (Table 1).

Quality assessment analysis of the EpiCurator

A robust and refined accurate selection of epitopes is crucial to improve the development of peptide-derived vaccines. To this goal, the EpiCurator brings together six analyses (Conservancy, Human homology, Epitopes overlap, EpiMiner, IEDB matching, and Mutation screening) to the accurately selected epitopes (Fig. 1B).

To the workflow’s quality assessment analysis, we sought to characterize the number of epitopes taken for the main analysis. The epitopes conservancy selected 29.23% of the predicted epitopes with 100% identity across at least 90% of the SARS-CoV-2 samples. Nevertheless, 70.77% of the predicted epitopes do not correspond to the conservancy parameter (Fig. 2, Fig. S2). Human homology analysis identified only 0.1% of the predicted epitopes sequence in the human genome, keeping 99% accurately selected, unmatched with human sequences (Fig. 2, Fig. S2). In addition, the screening by the previously published epitopes (EpiMiner and IEDB matching analysis) allowed selected > 80% of new epitopes, since 17.57% of them have already been described in the literature and/or IEDB server (Fig. 2, Fig. S2). To confirm the effectiveness of the EpiMiner we provide all the articles IDs and DOI in which the epitopes were found in Table S4.

Figure 2 Efficiency of EpiCurator analysis for accurate selection of epitopes.

The plot represents the percentage of epitopes removed in each different analysis.

To further assess the quality of the EpiCurator analysis, a pairwise comparison validation was performed. The comparison used NGS data reported by four papers with substantial similarity with our methodologies (Kiyotani et al., 2020; Chen et al., 2020; Crooke et al., 2020; Chukwudozie et al., 2021). The prediction step allows identifying the same HLA class I and HLA class II-restricted T cell epitopes. At the same time, the accurate selection provided by the EpiCurator gathered 51.1% of the paper’s epitopes (Table S5). The conservancy across SARS-CoV-2 genomes samples reached 95,6% of the paper’s epitopes, with only 4.4% not conserved by our parameters. The human homology step certifies the absence of the article’s selected epitopes in the Human genome since the analysis does not recognize the homology of any of them. Interestingly, the IEDB matching step identified 45.6% of the paper’s epitopes as having already been published on the IEDB server showing the importance of this step analysis if we want to describe the epitopes by the first time (Table S5).

Properties of accurately selected epitopes

Assembling the workflow analysis results, 199 (3.78%) HLA class I-restricted T cell epitopes were selected from SARS-CoV-2 proteins, mainly identified in ORF1ab (n = 154 (77.4%)) and Spike glycoprotein (n = 15 (7.5%)) (Table 2, Fig. 3A). The epitopes keep a high affinity for HLA-C alleles (Fig. 3C). The HLA class II epitopes’ accurate selection reached 153 (2%) epitopes, also mainly identified in ORF1ab (n = 111 (72.5%)) and Spike glycoprotein (n = 22 (14.4%)). The HLA class II epitopes showed an affinity prevalence for the HLA-DPA1-DPB1 haplotypes (Fig. 3D). The comprehensive workflow analysis for B cell epitopes selected 14.6% of predicted epitopes with 60% of the epitopes from the Spike glycoprotein (Table 2). Interestingly, all epitopes have high antigenicity (0.92 ± 0.30), are non-toxic and non-allergenic (Table S3).

Table 2 Summary of final SARS-CoV-2-derived epitopes accurately selected to each SARS-CoV-2 protein.

	HLA class I-restricted T cell epitopes	HLA class II-restricted T cell epitopes	B cell epitopes	
Protein	Prediction
(n)	Curation
(n - %)*	Prediction
(n)	Curation
(n - %)*	Prediction
(n)	Curation
(n - %)*	
Envelope	33	–	54	1 (1.85)	10	1 (10)	
Membrane	99	1 (1.01)	83	–	7	–	
Nucleocapsid	226	10 (4.25)	390	5 (1.28)	147	14 (9.52)	
ORF10	23	–	10	–	–	–	
ORF1ab	3621	154 (4.25)	5225	111 (2.12)	–	–	
ORF3a	343	11 (3.21)	357	4 (1.12)	–	–	
ORF6	32	–	47	–	–	–	
ORF7a	96	3 (3.12)	102	–	–	–	
ORF7b	15	–	3	–	–	–	
ORF8	98	5 (5.10)	139	10 (7.19)	–	–	
Spike	675	15 (2.22)	1239	22 (1.77)	93	21 (22.58)	
TOTAL	5261	199 (3.78)	7649	153 (2.00)	257	36 (14.01)	
Notes.

* Percentage related to number of predicted epitope per protein.

Figure 3 Identification of final SARS-CoV-2-derived HLA class I- and class II-binding epitopes.

(A) Distribution of the accurately selected epitopes in the SARS-CoV-2 proteins. Each color represents a distinct protein. (B) Distribution of the accurately selected epitopes in each current Brazilian lineage. Each color represents a distinct epitope type. (C) Distribution of epitopes in HLA class I alleles. The color degree represents the variation of the number of epitopes. (D) Distribution of epitopes in HLA class II alleles. The color degree represents the variation of the number of epitopes.

Furthermore, the T cell epitopes immunogenic potential was assessed to characterize the capability of inducing in silico protective immune responses. The analysis identified the profile of HLA class I and HLA class II epitopes respectively as follow: IL-4 inducer activity (26.6% and 23.8%), IFNγ production (22.1% and 39.7%), immunomodulatory activity (2% and 3.3%), and proinflammatory activity (78.9% and 81.4%) (Table S3).

In addition, to assess the capability to be an in silico Brazilian epitopes candidate, we sought to estimate their distribution among the Brazilian circulating lineages at the time of analysis (P.1, P.2 and B.1.1.28). Firstly, we identify the number of epitopes in common across P.1, P.2 and B.1.1.28 lineages identifying the highest proportion of HLA class II epitopes (61.4%), following by HLA class I (46.7%) and B cell (14.3%) (Fig. 3B). Thereafter, considering the most representative lineage in Brazil, at the time data was retrieved (P.1), remarkably, proportions highest than 50% for all the epitopes were identified, with 70.1% of HLA class II epitopes, 69.8% of HLA class I, and 51.4% of B cell epitopes in P.1 (Fig. 3B).

In the last analysis, considering the HLA promiscuity epitopes selection, the population coverage for the T cell epitopes associated with their respective HLA allele binding (Table S3) was estimated by the IEDB server. Notably, the T cell epitopes have a wide population coverage, presenting 99.52% (HLA class I epitopes) and 100% (HLA class II epitopes) of cumulative Brazilian population coverage and 98.45% (HLA class I epitopes) and 99.87% (HLA class II epitopes) of worldwide population coverage (Table S6).

Epitope-specific RBD Spike as a baseline for validation of EpiCurator selection

The EpiCurator allows the selection of the majority of the epitopes in the ORF1ab and Spike glycoprotein. The SARS-CoV-2 Spike glycoprotein has greater prominence concerning the virus and host interaction and has been the main target in epitopes prediction (Shang et al., 2020; Walls et al., 2020). Therefore, we assumed the spike epitopes as a baseline to validate the EpiCurator accurate selection. The Spike epitopes (n = 58) were mainly in the N-terminal domain (NTD) (53.4%) and receptor-binding domain (RBD) (18.9%) (Fig. 4).

The RBD epitopes (n = 11) comprise three of HLA class I (461-NYNYRYRLF-469, 506-QSYGFQPTY-514, 516-FGYQPYRVV-524), three of HLA class II (322-EKGIYQTSNFRVQPI-336, 322-EKGIYQTSNFRVQPR-336, 443-TGCVIAWNSKNLDSK-457), and five B cell epitopes (332-RVQPTES-338, 424-APGQTGK-430, 424-APGQTGT-430, 455-DSKVGGNYN-463, 474-LKPFERD-480) (Fig. 4).

To evaluate the robust and refined RBD epitopes selection, we perform a diversified analysis. They unveil the highest antigenicity (1.1 ± 0.27) (Fig. 5A) reflecting their ability to bind molecules of adaptive immunity. Notably, their affinity for several HLA alleles are high (0.49 ± 0.15) (Fig. 5A) and achieves promiscuity with ≥ 5 HLA allele binding per epitope (Fig. 5B). The RBD epitopes HLA data pairwise with population coverage that remains at around 80% in the Brazilian and worldwide population (Fig. 5B). To further characterize the RBD epitopes, they reach the conservancy of over 99% across SARS-CoV-2 genomes samples of lineages P.1, P.2 and B.1.1.28 published on GISAID (See Methods) (Table S3).

Figure 4 Distribution of the accurate selected epitopes in the structure of SARS-CoV-2 Spike glycoprotein.

Representation of Spike protein structure and their main domains. The epitopes are distributed in the Spike structure by their coordinate in the protein sequence allowing the identification of the domains where the epitopes were found. The specific sequence and coordinate of the epitopes found in the RBD domain are shown. The epitopes are colored by their type (Class I, Class II and B cell epitopes).

The high conservancy observed suggests that epitope sequences are shared in the circulating Brazilian lineages. Considering the natural homology among the SARS-CoV-2 lineages but regardless of the epitopes’ conservancy with other lineages, we expand the identification of the RBD epitopes for diversified samples available in GISAID at the time of writing. Interestingly these epitopes were identified in more than 1 million samples for around 1 thousand lineages of GISAID (Table S7). These findings include some samples of current VOCs: Alpha (B.1.1.7), Beta (B.1.351), and Delta (B.1.617.2) (Table S7). Concern about the natural lineages’ homology mentioned, we perform a comparison among spike glycoprotein of described Brazilian lineages and VOCs. The evaluation of random 100 samples available in GISAID of each lineage shows 99.1% of identity with 99.9% of coverage confirming the sequence lineages similarity (Table S8).

One of the mentioned properties of RBD epitopes is the high affinity for HLA alleles. To validate these findings the in silico structural binding performance was assessed by the PepDock docking tool. Thus, the RBD epitopes and the most major HLA alleles in studies with COVID-19 (see Methods) were structurally bonded in complex HLA-epitope (Fig. S3). All complexes were seen with high-performance scores demonstrating in silico epitope effectiveness in activating the cell-mediated immune response via MHC presentation. Their high estimated accuracy selected the two most significant HLA-epitope complexes (Table S9, Fig. 6). The HLA class I epitope 516-FGYQPYRVV-524 in complex with HLA-A*01:01, HLA-A*02:01, HLA-B*08:01, and HLA-C*12:03; and the HLA class II epitope 443-TGCVIAWNSKNLDSK-457 in complex with HLA-DRB1*03:01 and the HLA-DRB1*12:02 alleles (Table S9, Fig. 6). Additionally, the best measured free energy of the complexes was ΔG = −23 kcal/mol for 461-NYNYRYRLF-469 HLA class I epitope in complex with HLA-A*01:01 and ΔG = −31.6 kcal/mol for 322-EKGIYQTSNFRVQPR-336 HLA-class II epitope in complex with the HLA-DRB1*04:01 allele (Table S9, Fig. 6). An additional analysis was performed to assess the specific residues involved in the structural binding allowing to identify the specific amino acids in contact between the epitopes and the HLA alleles (Table S9).

Figure 5 Properties of epitopes from Spike RBD domain.

(A) The affinity values of the selected epitopes (x axis) are indicated as bars on the right y axis, and the antigenicity values are indicated as dots on the left y axis. (B) The number of HLA binding alleles for the selected epitopes (x axis) are indicated as bars on the left y axis, and the cumulative percentage of population coverage is depicted as dots on the right y axis. Each color of bars represents a distinct epitope type and each color of lines represents a distinct population.

Multi-epitope construct for in silico validation of RBD epitopes

To further validate the RBD epitopes and consequently confirm the effectiveness of EpiCurator in providing a curated selection, the epitopes were inserted in a multi-epitope construct. It consists of 287 amino residues, including 11 RBD selected epitopes joined by linkers (Fig. S1). The structural appraisal of the secondary structure predicted by the PSIPRED server revealed 31.7% alpha-helix, 13.5% beta-strand and the disordered region was 12% (Fig. S4). Some evaluation of the linear sequence of multi-epitope were identified as follows: high antigenicity by the Vaxijen 2.0 (0.87) and the AntigenPro (0.93), good solubility by SolPro (0.91), Protein-sol (0.66), and SOLart (65.80%), non-allergenic feature by AllerTOP and AllergenFP, hydrophilic property (hydropathicity = −0.438) and stability (instability index = 20.56), with pI 7.03, and molecular weight calculated to be 29 kDa.

Another assessment of linear sequence includes the in silico immunogenic profile provided by the C-IMMSIM immune server. The analysis showed a gradual increase of IgM, IFN-γ (associated with both CD8+ T cell and CD4+ Th1 response as shown in Fig. 7A), and IL-2 level after each multi-epitope exposure indicated an elevated immune response (Fig. 7B). Besides, an adequate generation of both IgG1 and IgG2 was shown (Fig. 7C) with high levels of clonal proliferation of B cell and T cell population (Figs. S5A and S5B). In addition, the development of in silico immune memory was assessed by the abundance of different types of B cells and T cells (Figs. S5C and S5D).

Figure 6 Structure of the HLA-epitope complex for the main T cell epitopes in Spike RBD domain.

Structure complexes provided by docking simulation shows the MHC binding grooves (blue ribbons), and the epitope (red structure). For each complex, the amino acids sequence of the epitope, HLA binding allele and the binding free energy are available.

The favorable in silico immunogenic profile of linear construct led to a multi-epitope 3D modelling to assess TLR binding capability adding validation for the curated selection provided by the EpiCurator. In this context, we modeled and evaluated the 3D multi-epitope to choose the best model for the TLR binding assay. The best predicted tertiary structure returned by RaptorX has an RMSD score of 11.218 (Fig. S6A), 62% of residues are predicted to be exposed and 30% are predicted to be disordered (Fig. S6B). The refinement by GalaxyRefine allows the choice of the best model based on its quality scores available in Table S10. This model highlights the epitopes according to their type (B cell, HLA class I T cell and HLA class II T cell) (Fig. S7). The overall model quality of the refined structure was further validated in the ProSA web (Fig. S8A), showing the energy results by position (Fig. S8B) with good local quality since all energy values of the residues are negative.

Considering the accuracy of the refined model, a docking between our multi-epitope and the immune receptors (TLR3 and TLR4) was performed to check stability and binding affinity. The best complexes (Fig. 8) for each receptor had free energy values of ΔG = −1,048 kcal/mol (TLR3) and ΔG = −1020 kcal/mol (TLR4), indicating a high binding affinity. In addition, the stability and physical movements of the complexes were confirmed using molecular dynamics simulation in the iMODS server. The complex results are presented in Figs. S9 and S10.

Figure 7 In silico simulation of immune response using multi-epitope construct as an antigen.

The multi-epitopes injection occurs on day 1 and within a 6-month interval. (A) Evolution of Th0, Th1, Th2 and Th17 response in cell/mm3 and percentage across the days. (B) Cytokine production across the days - specific subclasses are indicated as colored peaks. (C) multi-epitope injections (black vertical lines) promoting the immunoglobulin production - specific subclasses are indicated as colored peaks.

Figure 8 Structure of the multi-epitope-TLR complexes.

The immune receptors TLR3 and TLR4 are demonstrated in gray color and the ligands (multi-epitope) are shown in pink color. (A) The multi-epitope-TLR3 complex (B) the multi-epitope-TLR4 complex.

Discussion

Several studies have used immunoinformatic approaches to select B cell and T cell SARS-CoV-2 epitopes for epitopes-based vaccine formulation (Ramana & Mehla, 2020; Oli et al., 2020; Siracusano, Pastori & Lopalco, 2020; Yoshida et al., 2021; Ribes, Chaccour & Moncunill, 2021; Jin et al., 2021). However, current computational methods are limited since they identify a large number of epitopes and need different web servers for accurate selection (Bui et al., 2006; Doytchinova & Flower, 2007b; Gupta et al., 2013). This study developed an approach that gathered diversified analysis assembled in a workflow (EpiCurator). It accurately selects SARS-CoV-2 epitopes with useful in silico properties for an immunogen candidate target.

Its effectiveness was validated with a pairwise comparison analysis with four previously published papers (Kiyotani et al., 2020; Chen et al., 2020; Crooke et al., 2020; Chukwudozie et al., 2021). The main analysis of this validation was the IEDB matching that promotes a robust and refined selection of epitopes which confirms the importance of the IEDB database in the epitopes analysis studies (Beaver, Bourne & Ponomarenko, 2007). In addition, our approach can be used for extensive scale search and high-throughput computing analysis with the SARS-CoV-2 genomes available, an advantage to analyze a pandemic emergent virus (Ojha et al., 2020; Minervina et al., 2021; Pham et al., 2021).

Remarkably, our workflow conducts a thorough and facilitated accurate selection that enables us to prioritize the epitopes candidates. The EpiCurator recognizes patterns of cross-conservation with SARS-CoV-2 and human epitopes, eliminating the ones with significant human homology (Meyers et al., 2021). In addition, notably identifying epitopes in published articles, ensuring the selection in silico candidates with a plausible assumption to be described by the first time in this study. Consequently, this analysis might also validate the effectiveness of our workflow’s accurate selection analysis, since our epitopes were identified in different studies previously published (Prachar et al., 2020). The workflow also allows the selection of epitopes by high conservancy (≥90%), across more than two thousand SARS-CoV-2 circulating Brazilian lineages in accordance with previous reports for SARS-CoV-2 epitopes selection (Zaheer et al., 2020; Mahapatra et al., 2020; Mallajosyula et al., 2021).

Three main phases comprise our approach to SARS-CoV-2 epitopes identification, the prediction, accurate selection and epitopes validation. Therefore, taking the spike glycoprotein epitopes as parameters, they represent 10% of the predicted epitopes and 2% of the accurately selected similarly to those presented in other reports (Kiyotani et al., 2020; Crooke et al., 2020) with the resembling result for the other proteins. Indeed, the spike data give us an advantage since they were used to optimize the validation of the epitopes accurately selected by EpiCurator. Thus, they are taken as a baseline to conduct a thorough analysis. This baseline is reached since the Spike plays the most crucial role in the entry of viral particles into host cells, promoting an effective infection (Ou et al., 2020). These characteristics make the spike a chosen target for epitope screening leading to vaccine development (Shang et al., 2020; Walls et al., 2020; Lin et al., 2020). Pertaining to the Spike, the RBD region induces responses that block S protein binding with the human cell receptor, neutralizing SARS-CoV-2 infection (Rakib et al., 2020; Yang et al., 2020), having substantial importance to peptide-based vaccine studies. Considering this importance, we prioritize 11 epitopes of this study identified on Spike RBD regions.

The RBD epitopes were identified in several SARS-CoV-2 lineages emphasizing samples of the current VOCs. They have favorable in silico structure interactions with the main HLA alleles related to the COVID-19 response (Tavasolian et al., 2020; Tomita et al., 2020; Shkurnikov et al., 2021). Thereby, it is plausible to assume that these epitopes could be responsible for the strong activation of the cell-mediated immune response (Patronov et al., 2011; Sarma, Olotu & Soliman, 2021) in case of an experimental assay. In addition, the RBD epitopes have a broad populational coverage consistent with studies for SARS-CoV-2 epitopes identified in lineages isolated from India, England and the United States (Mallavarpu Ambrose et al., 2021). These findings combined with the high conservancy, antigenicity and immunogenicity suggest these epitopes’ in silico immunogen profile (Zheng & Song, 2020; Mallavarpu Ambrose et al., 2021; Jahangirian et al., 2021), validating the accurate selection of EpiCurator.

Indeed, the RBD epitopes have substantial evidence and are seen as good epitope candidates. Despite sharing this evidence, they could together increase the immune responses and confer better protection against SARS-CoV-2 (Jahangirian et al., 2021). Accepting this assumption, they were used to design a multi-epitope construct to further validate their immunogenic properties (Kalita et al., 2020; Mohammad et al., 2020; Singh et al., 2020; Yang, Bogdan & Nazarian, 2021; Lim et al., 2021; Sharma et al., 2021). The multi-epitope has the capability of in silico activation of memory B and T cell with Th1 response. Several experimental studies about immune response against COVID-19 endorse these findings (Lipsitch et al., 2020; Hartley et al., 2020; Ghazavi et al., 2021; Quast & Tarlinton, 2021; Tarke et al., 2021). On the other hand, no response Th17 was found in the in silico assay, a common response that characterizes the severe COVID-19 profile (Wu & Yang, 2020). Additionally, the multi-epitope had low binding energy and high stability of in silico structure interactions with TLR3 and TLR4 similar to other computational studies (Kar et al., 2020; Rahman et al., 2020; Yang, Bogdan & Nazarian, 2021; Nemati et al., 2021; Saba et al., 2021). This interaction is important since TLR is an innate immune receptor that recognizes viral proteins and triggers infection resistance (O’Neill, Golenbock & Bowie, 2013). Furthermore, TLR3 and TLR4 are specifically related by different studies as part of immune response in COVID-19 (Patra, Chandra Das & Mukherjee, 2021; Khanmohammadi & Rezaei, 2021; Kaushik, Bhandari & Kuhad, 2021). The findings in multi-epitope analysis confirm the importance of the RBD epitopes, selected by the EpiCurator, as good in silico immunogens candidates, reinforcing our approach’s efficiency for a curated selection of epitopes.

Conclusions

It is important to reinforce that this work focuses on a computational prediction and selection of epitopes seeking to perform several in silico validation. Therefore, our approach is useful to researchers designing an experimental peptide-based vaccine to control the disease. Since, indeed, the development of an effective vaccine requires a detailed experimental investigation of the immunological correlations with SARS-CoV-2. Regarding the computational analysis, assembling the accurate selection of epitopes and their validation consolidate our approach as a helpful workflow analysis to provide epitopes for immunogens. These epitopes could significantly activate the in silico immune response against some circulating Brazilian variants. Furthermore, considering that the pandemic is still ongoing, our approach could contribute to continuous monitoring and identification of new SARS-CoV-2 epitopes according to the emergence of variants over time.

Supplemental Information

Supplemental Information 1 Schematic Presentation of the final multi-epitope construct

The multi-epitope structure is constructed by 11 subunits (blue subunits represents HLA class I epitopes, purple subunits represents HLA class II epitopes and yellow subunits represents B-cell epitopes), an adjuvant represented in orange, linked by EAAAK, AAY and GPGPG linkers (right legend). The linear sequence of the multi-epitopes is represented with the same colors previously listed, added with a histidine hexamer.

Click here for additional data file.

Supplemental Information 2 Efficiency of EpiCurator analysis for accurate selection of epitopes

The plot represents the percentage of epitopes removed and each different analysis identified in the caption for the HLA class I epitopes (A), HLA class II epitopes (B) and B cell epitopes (C).

Click here for additional data file.

Supplemental Information 3 Structure of the HLA-epitope complex for the main T-cell epitopes in Spike RBD domain

Structure complexes provided by docking simulation show the MHC binding grooves (blue ribbons), and the epitope (red structure) for HLA class I alleles (A) and HLA class II alleles (B). For each complex, the amino acids sequence of the epitope and HLA binding allele are available.

Click here for additional data file.

Supplemental Information 4 Graphical Representation of the Secondary Structure of multi-epitope construct

The alpha helix residues are in pink square, the beta strand residues are in yellow square, the coil residues are in grey square. The disordered residues are in Blue border square and purple border square and the epitopes are in black border square.

Click here for additional data file.

Supplemental Information 5 Cell activation under multi-epitope in silico immune Simulation

(A) Concentration of B cell population, including memory B cell after multi-epitope exposures. (B) Concentration of T-cell population, including memory cell after multi-epitope exposures. (C) Concentration of B cell population per state - Active, Internalized the Ag, Presentation on MHC II, Duplicating in the mitotic cycle and Anergic (D) Concentration of T cell population per state - Active, Duplicating in the mitotic cycle, Anergic and Resting - not active. For all panels the specific subclasses are indicated as colored peaks.

Click here for additional data file.

Supplemental Information 6 Structural properties predicted by RaptorX

(A) shows the solvent accessibility of each residue along the structure, and the proportion of buried, medium exposed and high exposed residues. (B) represents the disorder degree of each residue.

Click here for additional data file.

Supplemental Information 7 Final 3D structure of the multi-epitope

The structure refined using the GalaxyRefine server had all the RBD epitopes highlighted following their type: yellow color refers to HLA class I T-cell epitopes, pink color corresponds to the HLA class II T-cell epitopes, and the B-cell epitopes are represented by the blue color.

Click here for additional data file.

Supplemental Information 8 Structure validation provided by the ProSA web server

(A) shows the z-score comparing our multi-epitope structure to determined structures from PDB of the same size. (B) corresponds to the level of energy of each residue.

Click here for additional data file.

Supplemental Information 9 Molecular dynamic results for the multi-epitope-TLR3 complex

(A) refers to the eigenvalue and the needed energy for structure deformation; (B) shows the covariance matrix representing the unrelated (white), correlated (red) and anti-correlated (blue) residues; (C) demonstrates the results of the elastic network model and the darker gray regions point to more rigid springs; plot (D) shows the main-chain deformability simulation; and (E) represents the uncertainty quantification for each residue through the b-factor values.

Click here for additional data file.

Supplemental Information 10 Molecular dynamic results for the multi-epitope-TLR4 complex

(A) refers to the eigenvalue and the needed energy for structure deformation; (B) shows the covariance matrix representing the unrelated (white), correlated (red) and anti-correlated (blue) residues; (C) demonstrates the results of the elastic network model and the darker gray regions point to more rigid springs; plot (D) shows the main-chain deformability simulation; and (E) represents the uncertainty quantification for each residue through the b-factor values.

Click here for additional data file.

Supplemental Information 11 SARS-CoV-2 samples from GISAID

Click here for additional data file.

Supplemental Information 12 HLA alleles and haplotypes frequent in Brazilian population

Click here for additional data file.

Supplemental Information 13 Final selected epitopes: Summary of properties

Click here for additional data file.

Supplemental Information 14 Articles selected by EpiMiner analysis

Click here for additional data file.

Supplemental Information 15 Estimated data from EpiCurator pairwise comparison validation using sequences reported by four previously published papers

Click here for additional data file.

Supplemental Information 16 Estimated population coverages of each accurately selected T cell epitope

Click here for additional data file.

Supplemental Information 17 Estimated number of samples and lineage where RBD epitopes were identified

Click here for additional data file.

Supplemental Information 18 Homology analysis for the Spike protein among the Brazilian lineages and other non-Brazilian VOCs

Click here for additional data file.

Supplemental Information 19 Performance parameters of HLA-epitope docking simulation

Click here for additional data file.

Supplemental Information 20 Quality scores for the initial structure and the five candidate models after the refinement process

The quality features are QDT-HA, RMSD, MolProbity, Clash score, Poor Rotamers and the Ramachandran plot score (Rama favored).

Click here for additional data file.

We would like to thank all the authors and the administrators of the GISAID and IEDB databases, allowing this study to be properly conducted. The authors acknowledge the National Laboratory for Scientific Computing (LNCC/MCTI, Brazil) for providing HPC resources of the SDumont supercomputer, which has contributed to the results of the research reported within this paper. URL: http://sdumont.lncc.br.

Additional Information and Declarations

Competing Interests

Author Contributions

Data Availability

Ana Tereza R. Vasconcelos is an Academic Editor for PeerJ.

Cristina S. Ferreira conceived and designed the experiments, performed the experiments, analyzed the data, prepared figures and/or tables, authored or reviewed drafts of the paper, and approved the final draft.

Yasmmin C. Martins conceived and designed the experiments, performed the experiments, authored or reviewed drafts of the paper, and approved the final draft.

Rangel Celso Souza analyzed the data, prepared figures and/or tables, and approved the final draft.

Ana Tereza R. Vasconcelos conceived and designed the experiments, authored or reviewed drafts of the paper, and approved the final draft.

The following information was supplied regarding data availability:

The data is available at GitHub: https://github.com/YasCoMa/EpiCurator.

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
