# Peer review of "EpiCurator: an immunoinformatic workflow to predict and prioritize SARS-CoV-2 epitopes"

_PeerJ, doi:10.7717/peerj.12548_

## Round 0.1 · original submission · Major Revisions

Before a decision can be made on whether or not your manuscript is suitable for publication, a number of concerns need to be addressed as detailed in the reviewer comments below. Please pay attention to all of their comments in your revised manuscript. But in particular, you should have the manuscript carefully and thoroughly edited to improve grammar. In addition, you should more thoroughly delineate what distinguishes your work from previously published work on the same topic. In particular, you should explain how your work and results differs from that published by the investigators responsible for the IEDB web site.

I also want to add that you do not necessarily need to add the references indicated by reviewer 2. Doing so is entirely at your discretion.

Reviewer 1 ·

Basic reporting

The authors have reported a curation workflow for selecting epitopes with desired immunity. The manuscript sticks to the theme and is easily understandable.

The authors have cited most of the tools that they have used, except a few times. Like in the case of IEDB, authors have mentioned that they have used IEDB version 2.25. Though, the most recent version was published in 2018, and in 2019.
This is important in this case, because the older will surely have no epitopes from SARS-CoV2, which was discovered only in 2020.

In the table 2, it is very surprising that out of so many epitopes reported in the literature, the proposed approach failed to any of the epitopes in envelop protein and captured only 15 in the spike proteins. While we know that these proteins are the most immunodominant and therefore chosen for many vaccines running successfully.

Experimental design

The design follows the standard practices for epitope selections. It is impressive to see that the authors have been very transparent in describing each method alongwith the threshold/model used for prediction.

While performing conservation analysis, the authors relied on Blast, which might not be perfect tool, because of its design. The Blast is designed to search local alignment and the authors seem to apply it for global coverage.

Though at the end of the manuscript, I was expecting to see a table with final filters and sequences, which is missing.

Validity of the findings

In terms of validity, the authors have used a set of tools, and no experimental validation is done.
Another validation is that the breadth and length of epitopes were recently published to describe a bigger picture of the immune response (doi: 10.1016/j.chom.2021.05.010) was not reflected in their analysis.
There are many such reports published and authors have not shed any light on the comparison of those studies against theirs.

Reviewer 2 ·

Basic reporting

Authors have aimed to develop the workflow to predict the epitopes of SARS-CoV-2. According to the obtained results, a lot of epitopes were predicted and some of them were concluded to detect for the first time. In my opinion, there are some shortcomings of this study.

In this study, NetMHCpan and NetMHCIIpan were used to predict MHC-1 and MHC-2 epitopes (Figure 1). Both of them already work under the IEDB database and this database has been frequently used in a lot of studies predicting epitopes of SARS-CoV-2 until now. Depending on this, it is not clear how these new epitopes were predicted although limited alleles have been used. In my opinion, if authors examine more articles conducted in this field, I think that authors can recognize that these new epitopes have been detected before. Several articles that are not cited in this study are listed below with a limited search, and this shows that more investigations should have been done. As a result, this study is not suitable for publication yet.

- https://www.nature.com/articles/s41467-020-16505-0

- https://www.nature.com/articles/s41598-020-79645-9

- https://www.nature.com/articles/s41598-020-73371-y

Experimental design

Experimental design lack of originality.

Validity of the findings

Findings are not novel completely

Additional comments

The manuscript is not valuably acceptable for publication

Reviewer 3 ·

Basic reporting

The manuscript titled “EpiCurator: an immunoinformatics workflow to predict and prioritize SARS-CoV-2 epitopes” introduces a pipeline that combines different specialized prediction tools into a single workflow. This pipeline is used for predicting SARS-CoV-2 epitopes, with a focus on the lineages observed in Brazil for the purpose of enabling vaccine design and development targeted at the Brazilian population. The manuscript makes predictions which can be experimentally verified by other groups once they are made publicly available.

While the core idea behind the method is decent, and the results of the analysis and corresponding data are sufficient and adequate, the reporting style and grammar are not clear and need major improvements. The manuscript needs to be re-written, using the help of a person who is fluent in English.

There are some studies done in the labs of Shane Crotty and Alessandro Sette, that have extensively studied the role of T-cell immune responses to SARS-CoV-2 infections and vaccination. Since, the manuscript describes about including T-cell epitopes, in addition to B-cell epitopes, including a short paragraph providing a background of T-cell immune responses to SARS-CoV-2 infections would be good. There are also several studies on neutralizing antibody responses to SARS-CoV-2 infection/vaccination that show the effect of the mutations observed in the variants of concern, on the immune response, and these also need to be mentioned in the introduction. A brief mention about a new nanoparticle-based vaccine developed at the Institute for Protein Design currently in Phase 3 clinical trials at South Korea, would also be nice to include.

Experimental design

More emphasis needs to be placed on how the vaccine designed through these approaches will offer an advantage over current SARS-CoV-2 vaccines. Additional explanation/description of current SARS-CoV-2 vaccines for a bit of background and context could also be included.

The methods are detailed and easy to understand (but again- the sentences need to be rephrased and grammar corrected). It is important to specify that the findings made here, once validated, can be used for designing a vaccine that is specific to the Brazilian population (and not a universal vaccine), against the variants currently circulating in Brazil, since the genome sequences used are from samples isolated from Brazil.

Additional details are needed in the Methods and Results section to describe how the epitopes that were found to be highly conserved in Brazilian variants were also searched in other lineages.

Describe how the PepDock server models the peptide structures and the HLA-peptide complexes, and generates binding free energy values

Validity of the findings

The findings made in this study are predictions and will have to be experimentally validated. The fact that the study is able to predict epitopes that are already experimentally validated or present in the Immune Epitope Database reflects the reliability of the predictions.

---

## Round 0.2 · Minor Revisions

Thank-you for your resubmission of a much-improved manuscript. We will be happy to accept your manuscript for publication after a few minor revisions are made. As indicated by the review below, this entails providing current citations for the IEDB and GISAID databases and web sites.

Reviewer 1 ·

Basic reporting

The authors have improved on language and data presentation. I believe that in terms of references, authors have missed out on some of the most important references in their manuscript, like IEDB (database paper or analysis resource paper), GISAID are used a number of times, but not cited properly.

Experimental design

The authors have stated clearly that the manuscript is of computational profile and so is the design or validation.

Validity of the findings

The authors have provided data and also compared their analysis with the ones done previously.

---

## Round 0.3 · accepted · Accept

Thank-you for your revised manuscript. You have addressed all concerns, and I am now happy to accept your manuscript for publication.